# Effects of Cryoprotective Medium Composition, Dilution Ratio, and Freezing Rates on Spotted Halibut (*Verasper variegatus*) Sperm Cryopreservation

**DOI:** 10.3390/ani10112153

**Published:** 2020-11-19

**Authors:** Irfan Zidni, Yun Ho Lee, Jung Yeol Park, Hyo Bin Lee, Jun Wook Hur, Han Kyu Lim

**Affiliations:** 1Department of Marine and Fisheries Resources, Mokpo National University, Mokpo 58554, Korea; i.zidni@unpad.ac.id (I.Z.); hodj66@naver.com (Y.H.L.); jungyeol89@naver.com (J.Y.P.); leeehyobin@gmail.com (H.B.L.); 2Faculty of Marine Applied Biosciences, Kunsan National University, 558 Daehak-ro, Gunsan, Jeonbuk 54150, Korea; junwhur@kunsan.ac.kr

**Keywords:** cryodamage, cryoprotective agent, diluent, sperm motility, sperm survival rate

## Abstract

**Simple Summary:**

The spotted halibut, *Verasper variegatus*, is a popular fish species occurring naturally in the East China Sea and coastal areas of Korea and Japan. However, when reared in captivity, male and female spotted halibut do not usually mature synchronously. Maintaining production of this commercial fish in hatcheries through sperm cryopreservation is important. This study investigated the effect of several factors for successful cryopreservation of fish sperm including cryoprotective agents (CPAs), diluents, dilution ratios (Milt: CPA + diluents), and freezing rates. The observed factors significantly affected movable sperm ratio (MSR), sperm activity index (SAI), survival rate, and DNA damage after cryopreservation. In the present study, the mixture of 15% dimethyl sulfoxide (DMSO) + 300 mM sucrose with a dilution ratio lower than 1:2 and a freezing rate slower than −5 °C/min provided the best treatment and reduced DNA damage. These results can contribute to the development of a cryopreservation protocol for spotted halibut hatcheries.

**Abstract:**

The spotted halibut is species that has a high potential market value in Korea, but the supply of seed is unstable because of the limited milt production of males. The objective of this research was to explore different aspects, such as CPAs, diluents, dilution ratio, and freezing rates, to develop an optimal sperm cryopreservation. The parameters assessed were movable sperm ratio, sperm activity index, survival rate, and DNA damage. The CPAs tested in this research were propylene glycol, dimethyl sulfoxide (DMSO), methanol, ethylene glycol, and glycerol. Different diluents, including 300 mM sucrose, 300 mM glucose, Stain’s solution, and Ringer’s solution, were investigated. The previous experiment showed that the optimal CPA for cryopreservation was DMSO with a concentration of 15% with 300 mM as diluent. To determine the effect of the dilution ratio, sperm was diluted to 1:1, 1:2, 1:10, 1:100, and 1:1000 with 300 mM sucrose containing DMSO at a final concentration of 15%. Lastly, the optimal freezing rate of the sperm was evaluated with four different freezing rates (−1, −5, −10, and −20 °C/min). Post-thaw sperm motility was higher with a dilution ratio lower than 1:2, and the freezing rate was less than −5 °C/min. In conclusion, these findings represent the development of a cryopreservation protocol for spotted halibut.

## 1. Introduction

In South Korea, seed production techniques for many marine fishes have been developed to support commercial production. One marine species with a high potential market value in Korea is spotted halibut (*Verasper variegatus*). This species occurs commonly in the East China Sea and coastal areas of Korea and Japan [1], lives in areas with muddy sand bottoms at depths of up to 100 m, has a body length of up to 60 cm, and weighs up to 4.0 kg. According to the Korea seafood market brief for 2019, spotted halibut is among the top ten seafood products and is the most popular seafood imported from outside Korea. This fish is popular due to the fact of its good taste and nutritional qualities [2]. Spotted halibut culture technology is developing rapidly, especially in Korea and China; this is due to the rapidly decreasing availability of fish resources in nature. However, when reared in captivity, male and female spotted halibut do not usually mature synchronously [3]. One important biotechnology technique in the aquaculture industry is sperm cryopreservation, which carries numerous benefits. Sperm cryopreservation in the fish hatchery is a useful technique for effective and easy broodstock management and artificial fertilization. Blaxter [4], in 1953, released the first paper on fish sperm cryopreservation, and since then, sperm has been cryopreserved for more than 200 marine fish and freshwater species [5]. Sperm cryopreservation has several advantages, including synchronization of gamete availability, preservation of genetic material, simplified sperm transportation, decreased disease transfer, and reduced maintenance costs for male broodstock. Furthermore, sperm cryopreservation is a useful solution for the lack of mature male fishes in hatchery production and could facilitate artificial fertilization [6].

Understanding the physiology of gametes and the biochemical activities that occur throughout milt collection, processing, cooling, and thawing is essential to the improvement of sperm cryopreservation techniques. One of the most important factors in sperm cryopreservation is the composition of the cryomedium, which includes cryoprotective agents (CPAs) and diluents. Cryoprotective agents protect cells against the damaging effects of freezing and thawing [7], while diluents act to preserve sperm cells in an immotile condition and support their metabolic pathways. Therefore, an ideal diluent for cryopreservation of the sperm of a particular species aligns with the seminal fluid characteristics of that specific species [8]. Another essential factor for obtaining high-quality spermatozoa through cryopreservation is the freezing rate, which can be regulated by a computer-controlled freezer, as it can have a major effect on frozen thawed motility [9]. Among other factors that affect sperm cryopreservation in fish, the dilution ratio is related to physical shock during dilution [10], equilibrium time is related to penetration of cells and equalization of concentrations inside and outside of the cells [11], and thawing temperature is associated with a fast and full recovery of the integrity and permeability of the membrane [1,12].

Provided the species-specific effects of these variables, when developing a sperm cryopreservation technique for spotted halibut, it is necessary to perform studies to determine the optimal types and concentrations of CPAs and diluents. During the cryopreservation period, the number and kind of CPA can affect spermatozoa’s physiological and metabolic structures in different ways. As undiluted sperm is not appropriate for long-term storage; the species-specific cryopreservation technique requires a good diluent and CPAs [13]. A good cryoprotective agent is key to the successful cryopreservation of sperm of flatfish [14,15,16]. Previous studies have been performed on the development of effective strategies for cryopreservation related to the effects of cryopreservation diluents and CPA [17,18,19,20]. Despite this progress, frozen–thawed sperm quality frequently differed within individual batches, from batch to batch, and from male to male. Therefore, to increase the level of accuracy and reliability of a method performed by prior studies, other parameters, including DNA damage, need to be checked. Damage to cryopreservation-induced spotted halibut sperm DNA has not been evaluated to date. The goals of this experiment was to develop an optimized sperm cryopreservation protocol, including CPA, diluents, freezing rates, and dilution ratios, which are key factors for the successful cryopreservation of this species.

## 2. Materials and Methods 

### 2.1. Ethics Statement

The ethics committee for The Institutional Animal Care and Use Committee of Mokpo National University No.1183 (17 December 2013) allowed this research.

### 2.2. Broodstock Management and Collecting Sperm

The experiment was conducted from October 2019 until February 2020. Seven male spotted halibut with an average length of 38.03 ± 0.25 cm and weight of 372.2 ± 9.4 g were maintained in seawater flow, varying in temperature from 10–13 °C, with a salinity of 31.1 ± 1.3 psu, dissolved oxygen of 7.9 mg/L, and pH of 7.9 in a fisheries hatchery located in Yeosu-si, Jeollanam-do, South Korea. Spermiation was stimulated through intramuscular injection of Ovaprime (Syndel, Canada) upon a dose of 0.3 mL/kg to induce placement 2 days prior to collection of milt. To collect milt, fish samples were anesthetized with 50 ppm 2-phenoxyethanol, and sperm samples were acquired through stripping and immediately collected with 1.0 mL plastic syringes. Special care was taken to prevent contamination of sperm with seawater, urine, blood, or feces. Sperm samples were stored in an icebox. Samples of spermatozoa with overall motility greater than 80% and a sperm activity index (SAI) greater than 5 were collected for use in this experiment. The amount of sperm was 0.25–0.5 mL/fish with a sperm count of approximately 300,000,000~350,000,000 cells/mL.

### 2.3. CPAs and Diluents

Propylene glycol (PG), dimethyl sulfoxide (DMSO), methanol (MeOH), ethylene glycol (EG), and glycerol (G) were evaluated to test the influence of the CPA type on sperm cryopreservation in the first experiment. In this study, 300 mM of glucose and sucrose were applied as diluents, adding to the sperm sample each combination of five CPAs and two diluents to achieve a final CPA concentration of 10%. Each diluent and CPA mixture were collected in a 1.5 mL tube and applied at a 1:1 ratio to the sperm. No equilibration time was used. The mixture was loaded into 0.25 mL plastic straws (Sterile classical PETG sperm straw, India) and capped with a sealing powder (Reproduction Provisions, USA). The straws were frozen immediately after sealing. For the first cooling stage, the treated sperm was located in liquid nitrogen vapor for 2 min and immersed in liquid nitrogen (196 °C) for the second cooling stage. Control treatments were prepared from fresh sperm stripped from the same males. For each treatment, at least three straws were produced for use in the experiment, and the straws were thawed in distilled water at 10.5 °C prior to use. A second experiment was carried to investigate the effect of the diluent, in which 300 mM sucrose (S), 300 mM glucose (G), Stain’s solution (SS), and Ringer’s solution (RS) were applied. The components of SS were NaCL 0.75 g, KCL 0.038 g, egg yolk 20 mL, C_6_H_12_O_6_ 0.10 g, and NaHC0_3_ 0.20 g in 100 mL H_2_O [21]. The RS contained NaCl 1.35 g, KCl 0.06 g, NaHC0_3_ 0.02 g, CaCl_2_ 0.025 g, and MgCl_2_ 0.035 g in 100 mL of distilled water [22]. For this experiment, 15% DMSO was used as the CPA. Using the same procedure as the first experiment, each mixture was frozen and thawed. To identify the best concentration of DMSO, the third experiment was carried out as it was established in the first experiment as the most efficient CPA for cryopreservation. The DMSO, at various concentrations (5%, 10%, 15%, and 20%), was tested with 300 mM S as the diluent. The freezing and thawing method was the same as in earlier experiment.

### 2.4. Dilution Ratio and Freezing Rate

The previous experiment showed that the combination of 15% DMSO and 300 mM sucrose worked the best for the sperm cryopreservation in spotted halibut. In the third experiment, to evaluate the effect of dilution ratio (sperm: CPA + diluent), sperm was diluted to 1:1, 1:2, 1:10, 1:100, and 1:1000 with 300 mM sucrose containing DMSO at a final concentration of 15. Lastly, to determine the optimal freezing rate, the sperm was evaluated using a controlled rate freezer (CRF; 14S, SY-LAB, Neupurkersdorf, Austria) with four different freezing rates (−1, −5, −10, and −20 °C/min). In this experiment the controlled-rate freezer was set to start from 10 °C until −80 °C for the first freezing, and then stored the frozen sperm in liquid nitrogen.

### 2.5. Motility of Frozen–Thawed Sperm and Survival Rate 

The movable sperm ratio (MSR) was measured as the ratio of the amount of moving sperm to the total amount of fresh or post-thawed sperm under a 200× microscope (CH30, Olympus, Tokyo, Japan). The SAI was calculated using the formula (index score × MSR (%))/100, with the index score acquired from scoring of sperm depend on their performance [23]. Sperm activation was conducted using filtered seawater with a ratio of 1:100 at a temperature of 25 °C. The thawed milt was mixed with filtered seawater, and MSR and SAI were observed through microscopy within 30 s of dilution.

The Cell Counting Kit-8 (Bimake, Houston, TX, USA) was used to calculate the survival rate of frozen—thawed sperm, and the absorbance was read using a microplate reader at a wavelength of 450 nm (Spectra Max 190, Molecular Devices, San Jose, CA, USA). In the first stage, the cell suspension (100 μL/well) was inoculated in 96 well plates. The plates are incubated using an incubator with humidification (37 °C, 5% CO_2_) before use. Into each well of the plate, 10 μL of the kit solution was added and then incubated for 1–4 h. Absorbance was measured using a microplate reader at 450 nm. According to Bimake CCK-8, the sperm survival rate determined by counting the amount of formazan (orange dye) provided by dehydrogenase activity in cells [24]. Live sperm is indicated by an orange hue, while white shows dead sperm cells. The survival rate of frozen–thawed sperm was represented as the fresh sperm percentage survival, the abundance of which was taken as 100%.

### 2.6. Single-Cell Gel Electrophoresis Assay

Using the best mixture of CPA and diluent chosen in earlier studies, a single-cell gel electrophoresis assay (CometAssay, Trevigen, Gaithersburg, MD, USA) was used to evaluate DNA damage to sperm during the freezing and thawing period. For detection under a fluorescence microscope (DM 2500, Leica, Wetzlar, Germany), the sperm was then colored for 30 min in SYBR Gold (diluted 10,000 in DMSO). The number of tail intensity, head intensity (DNA_h_), whole comet area (DNA_c_), and tail migration were accessed by the Comet Assay IV Lite method (Instem, Staffordshire, UK). Using the formula (100 × (DNA_c_–DNA_h_)/DNA_c_), the percentage of tail DNA (DNA_t_) was measured [23,24,25]. The DNAc and DNAh are the sum of the intensities of the pixels in the whole comet area and in the head area, respectively [26]. Data were expressed as the average, and the percentage of cells within each different degree of damage was also established.

### 2.7. Statistical Analyses

All variables are presented as the mean ± SE, and previous to analysis, percentage data were standardized by arcsine square root transformation. Substantial differences between parameters were measured using variance analysis (ANOVA) and Duncan’s multiple range test (*p* > 0.05) given by SPSS software version 23. The correlations among all variables were checked using bivariate Pearson’s correlation coefficients.

## 3. Results

### 3.1. Effects of CPA and Diluent for Sperm Quality After Cryopreservation

We first investigated the effects of various CPAs for sperm cryopreservation of spotted halibut, testing PG, DMSO, EG, MeOH, and G. Using 300 mM G, the MSR was significantly higher for 10% DMSO, at 56.67 ± 1.67%, relative to other CPAs (*p* < 0.05). The SAI numbers were considerably greater for DMSO and PG, which each had a value of four compared with other CPAs (Figure 1). To assess the influences of different diluents for sperm cryopreservation in spotted halibut, we investigated 300 mM S, 300 mM G, SS, and RS with 10% DMSO (Figure 2).

The 10% DMSO + 300 mM S produced a significantly higher MSR (78.67 ± 0.58%) of frozen–thawed sperm relative to other diluents with no significant difference from the control (fresh sperm). A higher percentages of survival rate was shown in the control treatment followed by the 10% DMSO + 300 S (93.97 ± 1.30%) of post-thaw The SAI value showed no significant differences among diluents, as all had a value of five. As a diluent, RS produced lower values of both MSR and survival rate, followed by SS. The effect of DMSO concentration was investigated using 300 mM S as the diluent. Higher motility was observed in 15% DMSO (81.33 ± 0.88%), but it was not considerably different from that in 20% DMSO (76.66 ± 1.66%; *p* < 0.05). Treatment with DMSO also resulted in high SAI, but the survival rate of sperm was significantly higher in 15% DMSO (93.93 ± 1.32%) relative to all other concentrations (Figure 3).

### 3.2. Sperm Dilution Ratio and Freezing Rate effects on Frozen–thawed Sperm Motility

The effects of various dilution ratios of milt to the mixture (CPA + diluent) on the motility of frozen–thawed spotted halibut sperm are shown in Figure 4. The highest MSR, SAI, and survival rates were found for the ratios of 1:1 and 1:2, with MSR values of 80.67 ± 0.67% and 79.33 ± 1.20, an SAI values of five, and a survival rates of 95.09 ± 1.02% and 93.97 ± 2.29%, respectively, and there were no significant differences between these two treatments. On the other hand, for diluent ratios of 1:10 to 1:500, the MSR, SAI, and survival rates decreased significantly. Figure 5 shows the MSR, SAI, and survival rates of frozen–thawed sperm exposed to different freezing rates during the sperm freezing process. The highest MSR with a freezing rate of −1 °C/min was obtained (81.67 ± 1.67%), followed by a freezing rate of −5 °C/min (79.83 ± 1.04%), and there was no noticeable difference between these two methods. The high survival rate of frozen–thawed sperm also resulted from these freezing rates. The SAI values did not differ significantly differ among diluents, which all had a value of five.

### 3.3. Single-Cell Gel Electrophoresis

To identify any damaging effects of cryopreservation on DNA, we investigated various concentrations of DMSO using 300 mM S as the diluent. The highest values of head length and head intensity were found with 15% DMSO, at 51 ± 0.68 and 91.23 ± 0.67, respectively, which were higher than those obtained with other concentrations of DMSO as well as the control (fresh sperm). Treatment with 15% DMSO also demonstrated low levels of tail intensity, percent tail DNA, and tail migration relative to other treatments but had a higher level than the control of fresh sperm (*p* < 0.05) (Table 1).

### 3.4. Correlations among Frozen–thawed Sperm Motility, Survival Rate, and DNA Damage

The results of correlation analysis of frozen–thawed sperm motility and survival rate to DNA damage are shown in Figure 6. The percentage of tail DNA had a negative correlation to MSR (y = −7.6283x + 94.39, *R*^2^ = 0.9823), with more severe DNA damage alongside reduced sperm motility. Similarly, there was a negative association between the frozen–thawed survival rate and the percentage of tail DNA (y = −6.7196x + 106.16, *R*^2^ = 0.9137), with bigger damage of DNA associated with a lower frozen–thawed sperm survival rate. 

## 4. Discussion

In this work, the highest frozen–thawed sperm motility was observed using 15% DMSO, which is an effective CPA used in spotted halibut sperm cryopreservation. This result is similar to Tian et al. (2008) [1], who found 13.3% DMSO or PG to be successful CPAs, as no substantial differences in motility were evaluated between fresh and frozen–thawed sperm. The most effective CPA for fish spermatozoa has been identified to be DMSO [27], mainly because of its small molecular structure, which allows it to easily enter and exit the sperm cell [28]. The DMSO provided good frozen–thawed sperm quality in starry flounder [29,30,31]. For some other flatfish species, DMSO has been considered to be reliable, including Brazilian flounder (10% DMSO), summer flounder (10% to 20% DMSO), and stone flounder (7.5% to 10% DMSO), but not for others including winter flounder and yellowtail flounder [5,27,28,29]. These findings suggest strong species-specificity of CPAs. By contrast, glycerol and EG produced less than 5% motility after cryopreservation in this study. Under the same conditions, Tian et al. (2008) [1] used glycerol and EG and found that they produced low post-thaw sperm motility in spotted halibut. Use of sucrose-based diluents, including 300 mM S in the present study and TS-2 (110 mM sucrose, 100 mM KHCO^3^, 10 mM Tris-Cl, pH 8.2, osmotic pressure 335 mOsm/kg) used previously, showed similar results in the two studies. In the present study, glycerol was not suitable in combination with 300 mM S as a diluent, as it produced frozen–thawed sperm with low motility. However, different results were obtained by Liu et al. (2006) [2], who found that glycerol combined with egg yolk released the greatest motility, fertility, and hatching of frozen–thawed sperm. The motility of the frozen–thawed sperm appeared to increase with the addition of egg yolk in the results of Liu et al. (2006) [2]. Egg yolk was already tested as a diluent or CPA additive in many cryopreservation studies and has shown a positive effect. Considerably higher motility and viability values were identified in frozen milt of brown trout (*Salmo cettii*) in the presence of 10% egg yolk [32]. The existence of egg yolk in a 10% MeOH-containing diluent offered additional protection for salmonid sperm during freezing and thawing [33]. However, the advantageous effects of egg yolk addition tend to be species-specific, as increased fertilization rates have been identified with its use in Atlantic salmon (*Salmo salar*) [34] and rainbow trout (*Oncorhynchus mykiss*) [35], while equal or reduced fertilization rates have been identified in arctic char (*Salvelinus alpinus* L.) [36] and asp (*Aspius aspius*) [37]. Besides the benefits of using yolk on sperm cryopreservation, there are also weaknesses associated with protein sources from animals and critical wild population if overused [38]. Differences in the effectiveness of CPAs for a single species may occur due to the differences in the diluents combined with them. Moreover, various characteristics of the CPA and diluent impact their effects on the maintenance of sperm.

In addition to the CPA, diluents are an important factor affecting sperm preservation, as they are used for dilution of fish milt before cryopreservation and are commonly created to be compatible with the physiochemical structure of the seminal plasma of the fish [39]. In this investigation, high frozen–thawed sperm motilities were produced when 300 mM S was used as the diluent, which were significantly higher than the motilities obtained using 300 mM G, SS, and RS. During the cooling process, sugars like G and S preserve cellular membranes phospholipids [26,30]. The primary role of the diluent is to sustain spermatozoa in the immotile state until motility is necessary. Thus, the diluent plays an important function in cryopreservation. It is necessary in cryopreservation for various purposes, such as creating conditions which actively prevent the survival of sperm, avoiding the activation sperm due to the changes in osmotic pressure during dilution, and increasing the quantity of surviving sperm [6]. According to Muchlisin (2005) [40], 10% S (*w*/*v*) completely prevented structural and functional changes in sperm after short-term freezing. Sucrose has also been identified as an important diluent for fish sperm cryopreservation in other fish species including Brazilian flounder, winter flounder, summer flounder, spotted halibut, and Atlantic halibut [5,15,28,39,40,41,42].

When the dilution ratio approached 1:10 or higher, the motility of the frozen–thawed sperm decreased dramatically. In a previous study, cryopreservation of spotted halibut was applied with dilution ratios of 1:3 [2] and 1:2 [1]. Cryopreserved sperm of summer flounder showed higher viability with a dilution ratio of 1:2 [43]. For cryopreservation of Brazilian flounder and starry flounder, a dilution ratio of 1:3 has been effectively applied and dilution ratio between the 1:1 and 1:10 showed higher MSR in cryopreservation of stone flounder [16,18,29,37]. The duration of motility in Persian sturgeon (*Acipenser persicus*) sperm is highest with dilution ratios from 1:1 to 1:2 [44]. Although the optimal dilution ratio for cryopreserved fish sperm is species-specific and ranges from 1:1 to 1:20 [5], a higher dilution ratio generally reduces the percentage of motile sperm, SAI, and sperm survival rate during the dilution process. Fish sperm can be subjected to a rapid change in the local environment with a greater amount of mixture (CPA + diluent) than seminal plasma. In the current research, a significant impact of the dilution ratio on cryopreserved sperm motility was observed, indicating that a higher dilution ratio is correlated with lower frozen–thawed sperm motility. Similar to the findings of this study, more serious damage to sea bream sperm during freezing and thawing was caused by higher dilution [25]. Osmotic and physical stimuli to the sperm during dilution may explain the decrease in frozen–thawed sperm viability due to the high dilution ratio. This description is supported by the findings of Correa and Zavos (1995) [10], explained that the rapid osmotic pressure caused osmotic shock when diluents are added.

The optimal freezing/thawing rate differs greatly among species, such as Atlantic halibut and summer flounder, with its efficiency linked to the process of dehydration, protection of intracellular ice formation during freezing and recrystallization during thawing [15,20,44,45]. In the present study, the estimated optimal freezing rate was in the range −1 to −5 °C/min, indicating that these freezing rates were better able to cover sperm cells from intracellular ice crystal formation during the milt cryopreservation process than other rates tested. When freezing is slow, the cell may efflux intracellular water, causing dehydration [32]. Gwo (1994) [46] described cold shock sensitivity as being established by the content of membrane cholesterol and the balance of polyunsaturated fatty acids (PUFAs) which affect lipid fluidity. Differences in membrane composition of spermatozoa have strong effects on cell membrane adaptation to cold shock. The combination of 300 mM S as a diluent with 15% DMSO in this study appeared to alter the composition of the membrane that produces high MSR, SAI, and survival rates. 

One problem that can occur in the cryopreservation process is damage to sperm cell DNA. Identification of DNA damage is important for evaluating genetic issues in cryopreserved sperm and preventing the development of defective embryos from fertilized eggs [25]. In the present study, the lowest value of percentage tail DNA was found in the control treatment (fresh sperm) at 0.36 ± 0.06%, followed by treatment of 15% DMSO + 300 mM S (2.05 ± 0.16%), which showed a significant difference from other treatments with higher values. Significant DNA damage occurred in cryopreserved European sea bass (*Dicentrarchus labrax*) sperm compared with fresh sperm, but the cryopreservation process did not affect motility or fertilization percentage [47]. The DNA stability is only marginally affected by the cryopreservation of rainbow trout sperm, which does not affect the survival rate or quality of sperm [48]. Gwo and Arnold (1992) [49] reported no effect from the frozen–thawed process on the nucleus of Atlantic croaker spermatozoa. Although the sperm nucleus has been known to be secure and not affected by cryopreservation (Gwo 2000) [19], sperm cryopreservation in rainbow trout, European bass, and gilthead sea bream (*Sparus aurata*) was found to damage sperm DNA [16,18,47,50]. The level of damage to fish sperm DNA is affected by several factors, including the species of fish, its chromosome structure, and the CPA and diluent used for the process of freezing and thawing [20,30]. Based on the results of this study, cryopreserved sperm of spotted halibut exhibits greater DNA damage compared to fresh sperm, and the magnitude of this effect is inversely proportional to sperm motility and survival rate. According to Yavaş et al. (2014) [51], the methods of cryopreservation include the application of cryoprotectants, freezing, and thawing of sperm, which can result in damage to the spermatozoa and may reduce the rate of egg fertilization. As the results of the comet assay varied between DMSO concentrations, CPA was considered to be an important factor affecting DNA damage in spotted halibut after sperm motility. A variety of factors, such as formation of ice crystal within the cell, changes of pH, sudden immersion in cold water, osmotic shock, and CPA toxicity, affect cryopreserved sperm in a cumulative manner. Furthermore, DNA damage during cryopreservation is caused by the formation of reactive oxygen species that change DNA by oxidizing bases and creating breaks in the helix [51,52,53].

## 5. Conclusions

The optimal cryopreservation conditions in spotted halibut were found, and that using 15% DMSO as the CPA and 300 mM sucrose as the diluent resulted in the highest frozen–thawed sperm motility and a lower percentage of tail DNA. The freezing rates ranged from −1 to −5°C/min and had no significant differences for frozen–thawed sperm motility. Regarding the dilution ratio of sperm to CPA + diluent, ratios higher than 1:2 showed significantly lower frozen–thawed sperm motility.

## Figures and Tables

**Figure 1 animals-10-02153-f001:**
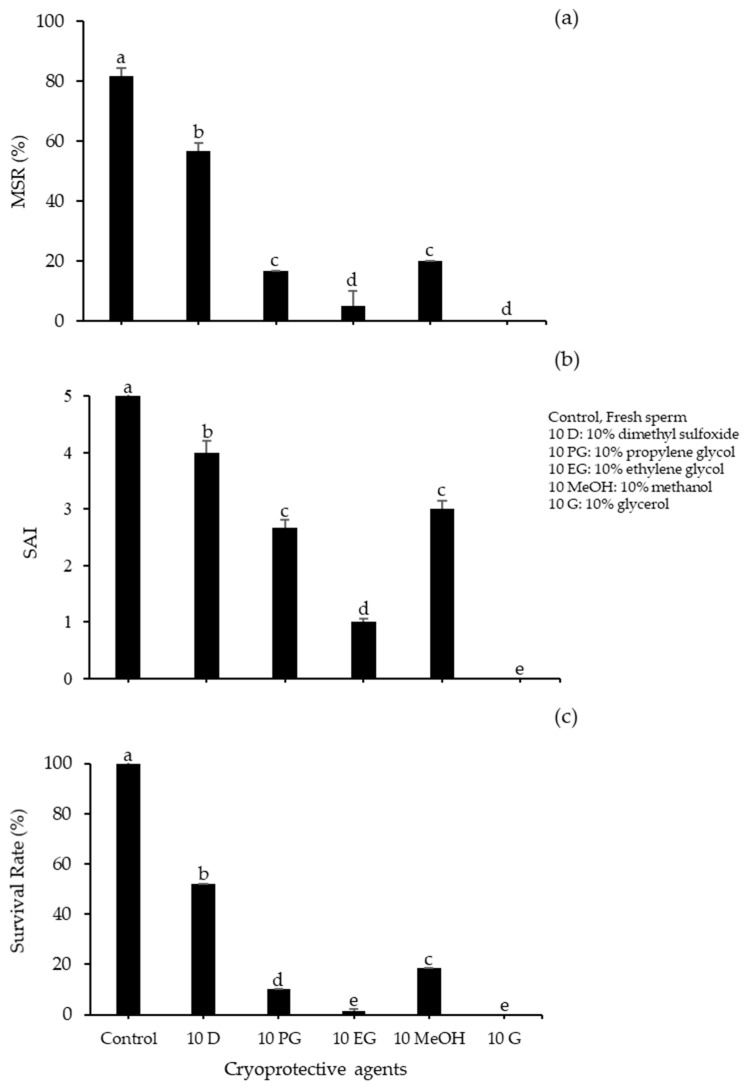
Performance of spotted halibut frozen–thawed sperm with 300 mM glucose at various cryoprotective agents (CPA)s: (**a**) movable sperm ratio (MSR); (**b**) sperm activity index (SAI); (**c**) survival rate. Significant differences between CPAs are indicated by the letters (*p* < 0.05).

**Figure 2 animals-10-02153-f002:**
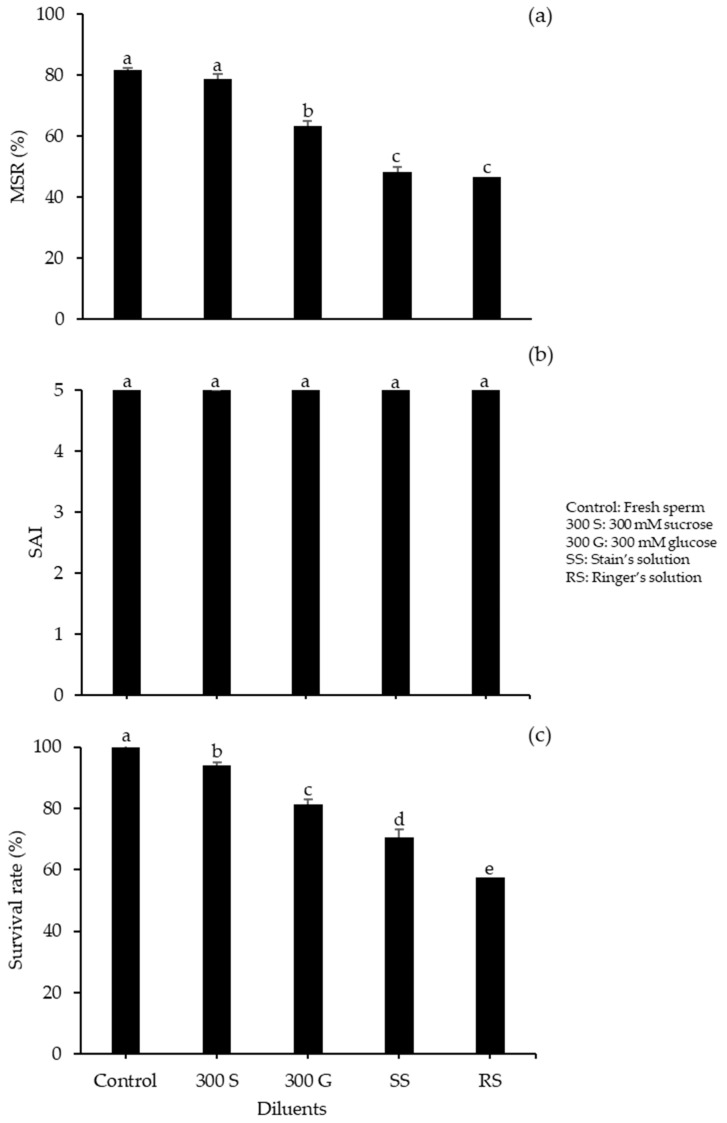
Performance of spotted halibut frozen–thawed sperm with 10% DMSO at various diluents: (**a**) MSR; (**b**) SAI; (**c**) survival rate. Significant variations between diluents are shown by small letters (*p* < 0.05).

**Figure 3 animals-10-02153-f003:**
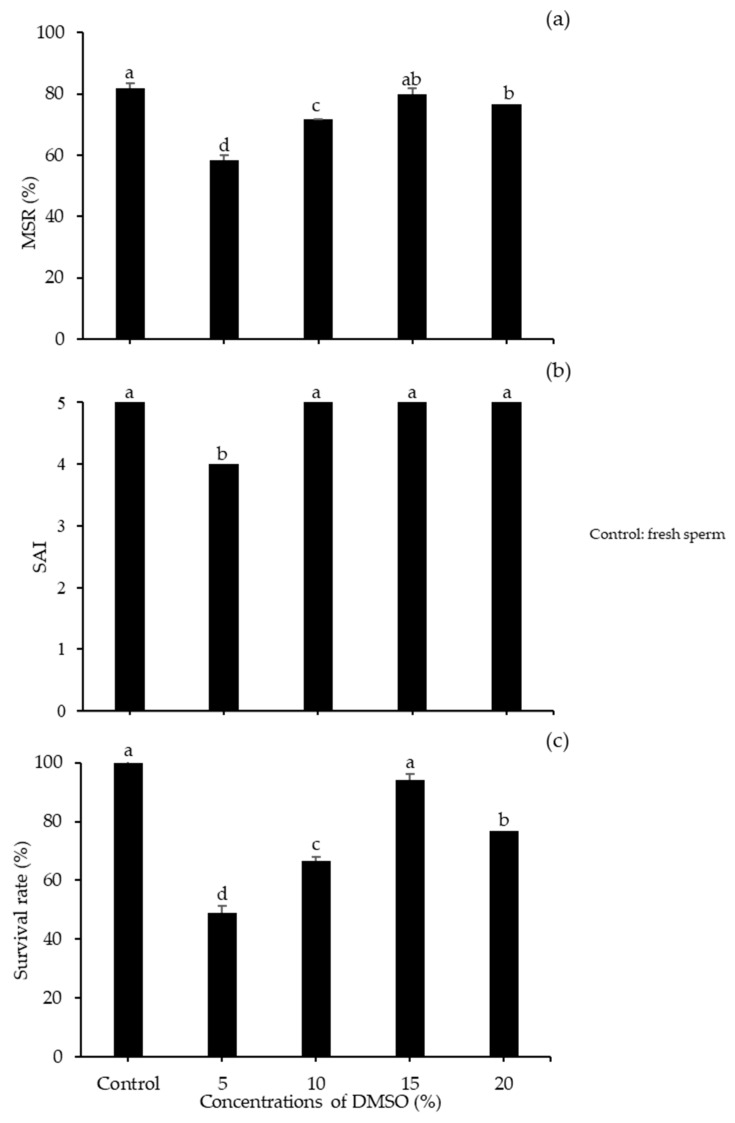
Performance of spotted halibut frozen–thawed sperm with 300 mM sucrose at various DMSO concentrations: (**a**) MSR; (**b**) SAI; (**c**) survival rate. Different letters indicate significant differences among concentrations of DMSO as a CPA (*p* < 0.05).

**Figure 4 animals-10-02153-f004:**
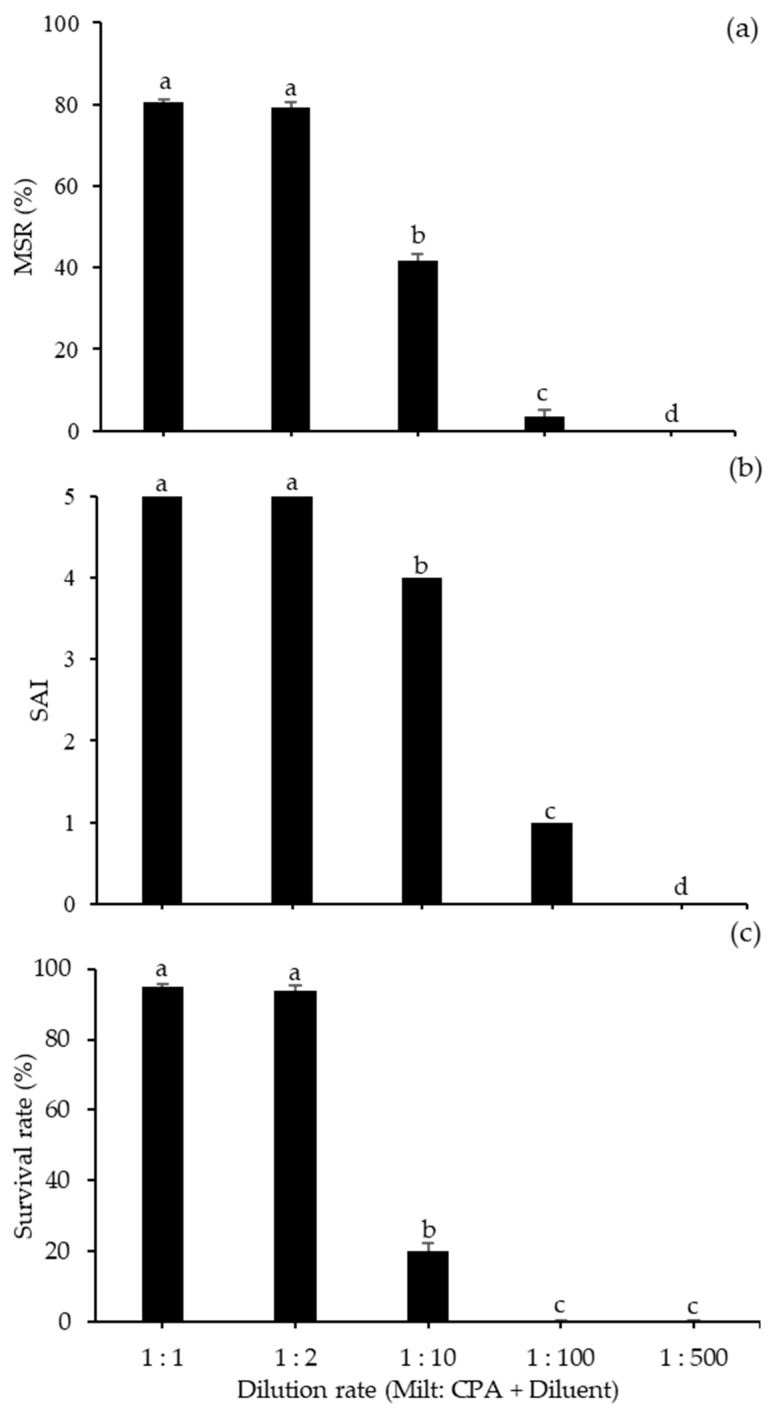
Performance of spotted halibut frozen–thawed sperm depending on the dilution rate (milt: CPA + diluent) used for sperm freezing: (**a**) MSR; (**b**) SAI; (**c**) survival rate. Important variations among dilution rates are suggested by different lowercase letters (*p* < 0.05).

**Figure 5 animals-10-02153-f005:**
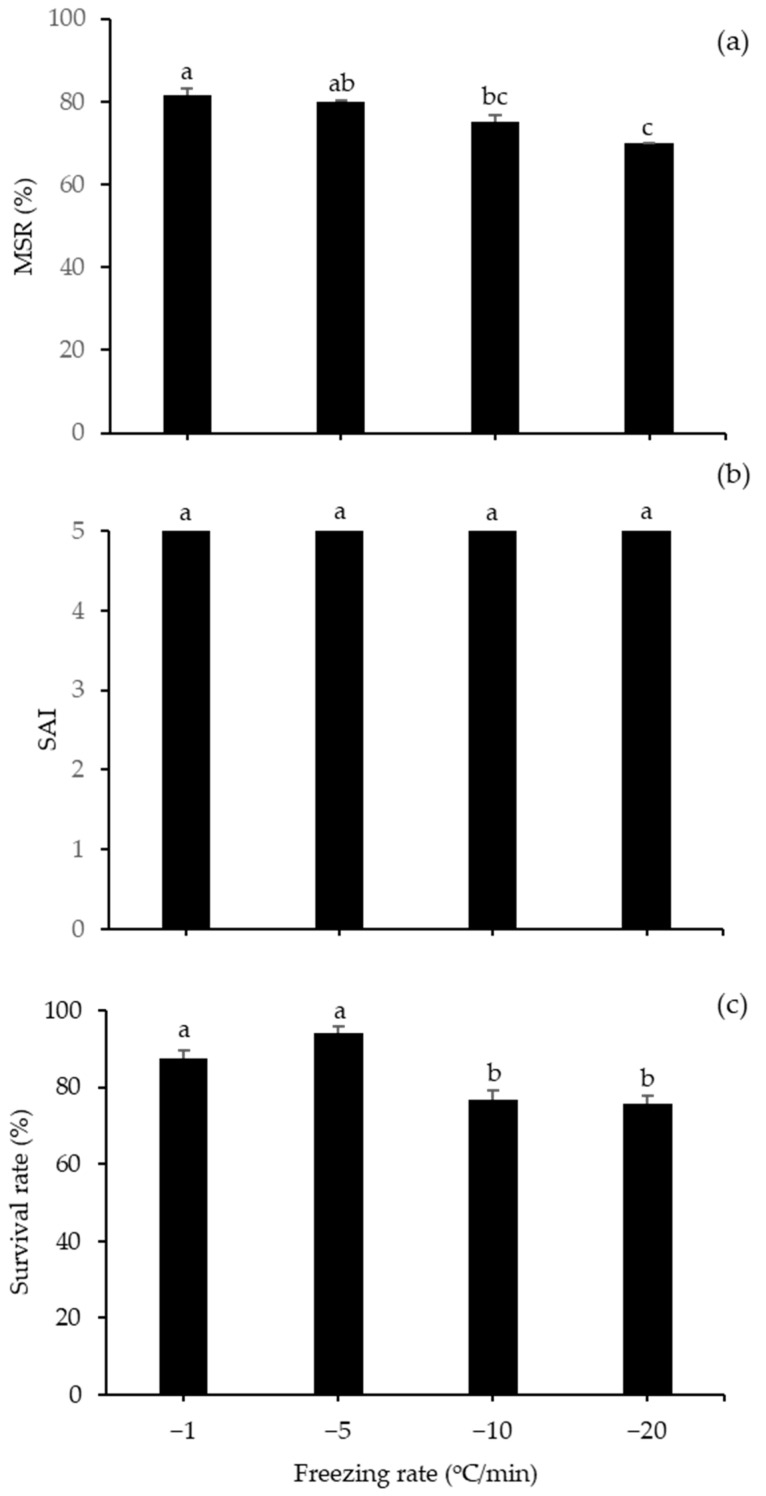
Performance of spotted halibut frozen–thawed sperm for various freezing rates (−1, −5, −10, and −20 °C/min) with 15% DMSO and 300 mM sucrose: (**a**) MSR; (**b**) SAI; (**c**) survival rate. Different lowercase letters suggested important variations among freezing rates (*p* < 0.05).

**Figure 6 animals-10-02153-f006:**
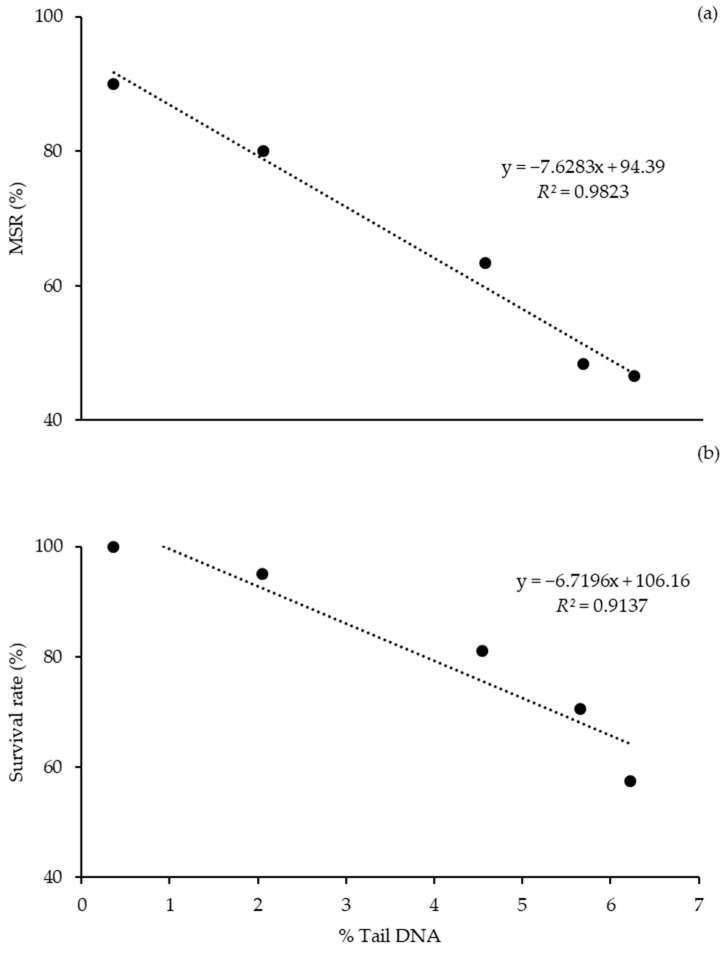
Correlation between MSR and survival rate to percentage of Tail DNA (x) in frozen–thawed sperm: (**a**) relationships between MSR and % Tail DNA; (**b**) relationships between survival rate and % Tail DNA. *R*^2^ = Pearson’s correlation coefficient.

**Table 1 animals-10-02153-t001:** Statistical analysis of the comet assay conducted at various concentrations of DMSO and 300 mM sucrose for spotted halibut (*Verasper variegatus*) frozen–thawed sperm.

	Control	DMSO Concentration (%)			
		5	10	15	20
Head Length	46.09 ± 0.69 ^c^	40.63 ± 1.15 ^e^	42.90 ± 0.93 ^d^	51 ± 0.68 ^a^	48.34 ± 0.84 ^b^
Tail Length	21.44 ± 0.38 ^d^	46.34 ± 1.62 ^c^	51.99 ± 1.81 ^b^	29.46 ± 0.4 ^b^	60.63 ± 1.27 ^a^
Head Intensity	98.08 ± 0.27 ^a^	77.52 ± 1.76 ^d^	81.72 ± 1.48 ^c^	91.23 ± 0.67 ^b^	74.38 ± 0.93 ^e^
Tail Intensity	1.92 ± 0.27 ^e^	22.48 ± 1.76 ^b^	18.27 ± 1.48 ^c^	8.77 ± 0.67 ^d^	24.95 ± 0.92 ^a^
% Tail DNA	0.36 ± 0.06 ^c^	5.65 ± 0.48 ^a^	4.54 ± 0.31 ^a^	2.05 ± 0.16 ^b^	6.22 ± 0.21 ^a^
Tail Migration	0.82 ± 0.15 ^e^	26.10 ± 1.69 ^c^	30.57 ± 1.9 ^b^	4.63 ± 0.39 ^d^	36.46 ± 1.30 ^a^

Important variations between DMSO concentrations are suggested by different superscripted letters. Control: fresh milt.

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
