# Peer review of "Effects of Cryoprotective Medium Composition, Dilution Ratio, and Freezing Rates on Spotted Halibut (Verasper variegatus) Sperm Cryopreservation"

_animals, 2020, doi:10.3390/ani10112153_

Round 1
Reviewer 1 Report
The MS by Zidni et al. tried to improve the sperm cryopreservation protocols for spotted Halibut. For that, different cryoprotectant, extenders and dilution and freezing rates were assessed. It presents interesting preliminary results and many researchers are trying to improve sperm cryopreservation in fish species.
The entire manuscript must be rewritten carefully. The are several format errors in the entire MS text. I have read the manuscript carefully and put my suggestions. However, I consider that only if the authors make a thorough revision and remodeling of the whole MS, it could be considered for publication, not without first revising it.
Title should be rewritten, please avoid expression such as “treatment methods”.
I encourage the authors that a graphical abstract (or at least a separate section “Experimental design”) be included, mainly after reading summary and abstract at the beginning of the MS. The three different experiments are not well explained (I was unable to find throughout the whole manuscript a clear description of the experimental design in any place) and it is confusing.
Summary and abstract should be rewritten (see my previous comment). In the summary the authors should provide general information and conclusions about their research. However, in the abstract the experimental design must be clearly explained as well as the main results and conclusions should be included. Please, reorganize and rewrite both sections.
Summary
Line 18: please define the acronym CPAs
Abstract:
Line 33: dilution ratio and freezing rates evaluated should be enumerated in a similar way that CPAs and extenders used.
Introduction:
This section should be reduced. There are many redundant information. For instance, there are no need to provide the name of each species. But mainly, the information about cryopreservation protocol and the factors which may influence on it, must be well unified, organized and reduced.
Materials and methods
Without a well experimental design I have problems to follow the MS. How many treatments you have? You have to provide this information before the figures in the results section, which in my case, is the only place where I was able to find this information.
The information about Control treatment is missing. Only in line 359 (Discussion section), the authors refer Control as fresh sperm.
It would be advisable to check the sperm quality after dilution and previously to cryopreservation. Could the author provide this information?
Line 113: extreme treatment?
Line 168: please define DNAc.
Results:
Line 177: please rewrite this sentence “for sperm quality after cryopreservation”.
Control treatment should be considered, so the authors must describe the results taking into account this treatment, otherwise it has no sense that Control be included in the study.
Discusion:
This section should be reduced, in a similar way than Introduction. First paragraph should be removed (this information has been included in the Introduction). The authors should start with a brief description of their results and then immediately with the discusion of them.
Please remove the reference about Figures in this section. The figures should be described in the results section.
Line 306: the authors should point out the importance of avoiding origin animal substances, such as egg yolk, in the extenders regardless animal species.
Line 376: this conclusion is not supported by the results of this study since the sperm DNA status was not assessed when the authors compared different CPAs, diluents neither dilution or thawing rates. Please correct it.
Conclusions
Please unify both first sentences.
Author Response
Manuscript ID: animals-979702
07 November 2020
Dear reviewers,
I would like to re-submit the attached manuscript entitled “Effects of Cryoprotective Medium Composition, Dilution Ratio, and Freezing Rates on Spotted Halibut (Verasper variegatus) Sperm Cryopreservation.The manuscript has been carefully rechecked and appropriate changes have been made in accordance with the reviewer suggestions. Thank you very much for your consideration, and we really appreciate the comments and have learned a lot. I hope that the revised manuscript could be considered for publication in Animals journal.
Sincerely yours,
Han Kyu Lim
Mokpo National University
Department of Marine and Fisheries Resources, Mokpo National University, 1666
Youngsan-ro, Muan Jeonnam, 58554, Republic of Korea
limhk@mokpo.ac.kr

Reviewer 2 Report
The manuscript entitled “Effects of Cryoprotective Medium Composition and Treatment Method on Spotted Halibut (Verasper variegatus) Sperm Cryopreservation” aimed to explore various factors, including Cryoprotective agents, diluents, dilution ratio, and freezing rates to develop an optimal sperm cryopreservation for V. variegatus. The MS provides important and novel information. Introduction provides the purpose of the work and its significance. The current state of the research field was well reported. Materials and methods section provides sufficient detail to allow others to replicate and build on published results. Results section provide a concise and precise description of the experimental results. Discussion is well structured. However, correction to minor text edition is necessary. I recommend that you have your manuscript professionally edited before submission or read by a native English-speaking. Thus, I recommended to publish this MS in Animals after minor revision.
Minor comment
- Simple summary: I recommended to rewrite it. This should be written for a lay audience, i.e., no technical terms without explanations.
- Lines 18, 20 and subsequent: abbreviations (CPA, DMSO) should be defined in parentheses the first time they appear in the text.
- Line 39: the keywords of the paper should not contain any words already in the title but can include abbreviated terms or location information not suitable for the title. Please rewrite.
- Lines 92-95: confusing, please rewrite.
- Line 133: replace “NaHCO3” instead of “NaHC03”
- Citation and references have to be checked before publication.
Author Response
Manuscript ID: animals-979702
07 November 2020
Dear reviewers,
We are submitting a revised version of our manuscript, titled “Effects of Cryoprotective Medium Composition, Dilution Ratio, and Freezing Rates on Spotted Halibut (Verasper variegatus) Sperm Cryopreservation”.
We would like to further express our sincere thanks for the helpful comments from the reviewers. We have revised the manuscript according to their comments. We hope that the revised manuscript will now meet the requirements for publication in Animals Journal.
Sincerely yours,
Han Kyu Lim
Mokpo National University
Department of Marine and Fisheries Resources, Mokpo National University, 1666
Youngsan-ro, Muan Jeonnam, 58554, Republic of Korea
limhk@mokpo.ac.kr

Reviewer 3 Report
Lines 16, 96-97 - What does it mean for .... "sperm cryopreservation protocol for all stages of spotted halibut."?
- What equipment did you use for freezing?
- Specify better how the thawing was done!
- Detail the technique more. Did you use the subjective assessment of motility?
- Specify what is SAI?
- I suggest that you put the treatment legends in all the graphics presented
- Line 232 - These experiments are not described in the methodology, how were they carried out? What equipment did you use? I cannot analyze these results!
Author Response
Manuscript ID: animals-979702
07 November 2020
Dear Reviewers,
Thank you for giving us the opportunity to improve and resubmit our manuscript “Effects of Cryoprotective Medium Composition, Dilution Ratio, and Freezing Rates on Spotted Halibut (Verasper variegatus) Sperm Cryopreservation. Please find enclosed the revised manuscript for further consideration.
The manuscript has been revised according to the comments raised by the reviewer to the best of our ability. We would like to thank the reviewer for the constructive and competent criticism, and we hope that our manuscript will be acceptable for publication.
Sincerely yours,
Han Kyu Lim
Mokpo National University
Department of Marine and Fisheries Resources, Mokpo National University, 1666
Youngsan-ro, Muan Jeonnam, 58554, Republic of Korea
limhk@mokpo.ac.kr

Reviewer 4 Report
Solid experimental work that is relevant, well executed, written and referenced. I suggest having a native speaker of English go through the manuscript before submission, since there are a few mistakes throughout.
Author Response
Manuscript ID: animals-979702
07 November 2020
Dear Reviewer,
We are submitting a revised version of our manuscript, titled “Effects of Cryoprotective Medium Composition, Dilution Ratio, and Freezing Rates on Spotted Halibut (Verasper variegatus) Sperm Cryopreservation”.
We greatly appreciated your comments and those of the reviewers. These comments have helped us to improve our manuscript considerably and we hope that the revised version of our manuscript is now acceptable for publication in Animals Journal.
Sincerely yours,
Han Kyu Lim
Mokpo National University
Department of Marine and Fisheries Resources, Mokpo National University, 1666
Youngsan-ro, Muan Jeonnam, 58554, Republic of Korea
limhk@mokpo.ac.kr

Round 2
Reviewer 1 Report
The MS has been improved compared to the first version. However, it still requires some improvement.
Please get the English checked by a good scientific English writer. Many sentences are incorrectly constructed or need correction.
Line 19: please explain what you mean with “sperm:” in the brackets.
Lines 49-50: please rewrite this sentence.
Line 128: please, delete this sentence.
Lines 128-131: please rewrite the aim of the study and unify both sentences.
Line 162: please remove “For”.
Line 190: I suggest to leave “movable sperm ratio”. This is Material and Mathod section so the acronym should be defined.
Line 355: please rewrite the sentence and include “in spotted halibut” after sperm motility.
Line 327: in which species?? Please include it.
Lines 461-465: conclusions should be rewritten. The authors should recommend tresholds for dilution ratio and freezing rate. Please avoid expression such as “preferred”.
Author Response
Manuscript ID: animals-979702
15 November 2020
Dear reviewers,
We would like to re-submit the attached manuscript entitled “Effects of Cryoprotective Medium Composition, Dilution Ratio, and Freezing Rates on Spotted Halibut (Verasper variegatus) Sperm Cryopreservation. We greatly appreciated your comments and those of the reviewers. We have modified the manuscript according to the suggestion from the reviewer. Improvement of sentences, grammar checking, modification the aim of the study, modification of conclusions, and new references has been added in the revised manuscript. We would like to further express our sincere thanks for the helpful comments from the reviewers. We hope that the revised manuscript could be considered for publication in Animals journal.
Sincerely yours,
Han Kyu Lim
Mokpo National University
Department of Marine and Fisheries Resources, Mokpo National University, 1666 Youngsan-ro, Muan Jeonnam, 58554, Republic of Korea
limhk@mokpo.ac.kr
